# Geothermal Plus Sunlight-Based Incubator for Sustainable Pig Production

**Shad Mahfuz** [1,2,†] **, Hong-Seok Mun** [1,3,†]**, Muhammad Ammar Dilawar** [1]**, Keiven Mark B. Ampode** [1,4]**, Veasna Chem** [1]**, Young-Hwa Kim** [5]**, Jong-Pil Moon** [6] **and Chul-Ju Yang** [1,7,*]

[1] Animal Nutrition and Feed Science Laboratory, Department of Animal Science and Technology, Sunchon National University, Suncheon 57922, Republic of Korea
[2] Department of Animal Nutrition, Sylhet Agricultural University, Sylhet 3100, Bangladesh
[3] Department of Multimedia Engineering, Sunchon National University, Suncheon 57922, Republic of Korea
[4] Department of Animal Science, College of Agriculture, Sultan Kudarat State University, Tacurong City 9800, Philippines
[5] Interdisciplinary Program in IT-Bio Convergence System (BK21 Plus), Chonnam National University, Gwangju 61186, Republic of Korea
[6] Rural Development Administration, Jeonju 54875, Republic of Korea
[7] Interdisciplinary Program in IT-Bio Convergence System (BK21 Plus), Sunchon National University, 255, Jungangno, Suncheon 57922, Republic of Korea
[*] Correspondence: yangcj@scnu.ac.kr; Tel.: +82-61-750-3235
[†] These authors contributed equally to this work.

**Abstract:** This experiment was conducted to assess the effects of a geothermal plus sunlight-based incubator on the growth performance, electricity uses and housing environment of piglets. A total of 20 piglets, average 7.7 ± 0.015 kg (mean ± std.) initial body weight, were randomly divided into two separated incubators: control (conventional incubator) and the geothermal plus sunlight-based heat pump (GS) incubator with 10 replicated piglets. The experimental duration was 8 weeks. Average daily weight gain, feed intake, electricity consumption, and house temperature, humidity, ammonia, and carbon dioxide concentration were measured on a weekly basis. There were no significant differences in the final body weight, average daily body weight gain (ADG), average daily feed intake (ADFI) and feed conversion ratio (FCR) between the incubators. The electricity consumption of the GS incubator was reduced by 120.95 kWh/head and the saving efficacy was about 64.76% that of the conventional incubator. The electricity cost was reduced by 3.26 USD and the ratio of feed cost to weigh gain was lower in the GS-based incubator. No significant differences were noted for the internal temperature and humidity between the incubators. The ammonia concentration and carbon dioxide concentration were significantly lower ($p < 0.05$) in the GS-based incubator than the control incubator. The geothermal plus sunlight-based incubator might be healthy and economic for the sustainable pig production.

**Keywords:** geothermal plus sunlight-based incubator; body weight; electricity uses; ammonia; carbon dioxide; piglets

## 1. Introduction

A renewable source of energy is very important in the livestock and agricultural sector to protect the natural environment [1]. The over-application of fossil fuels has not only increased the energy prices worldwide, but it also raises demands for alternative system [2]. To capture the carbon dioxide from the source of emissions is not feasible in most cases, and thus to find out an alternate system for neutralizing the carbon dioxide from the source is essential [3]. The global energy supply is mostly used for electricity generation, while the majority is derived from fossil fuels. Fossil fuels are harmful to the environment by emitting greenhouse gases and other pollutants that contribute to climate change [4]. It is very important to identify an alternate heating or cooling system

that can reduce the uses of electricity [5]. The application of alternative energy sources in agriculture is increasing for its sustainable nature, high production performance and low maintaining cost [6]. High prices for energy and harmful gases from fuel have become an important issue in livestock production [7]. It is essential to maintain a comfortable environment (temperature, humidity, $CO_2$, $NH_3$ etc.) in swine house for the optimum growth and good health of pigs. Among the climatic factors, the ambient temperature is one of the most predominant factors that could influence the growth performance of pigs [8]. Weaning is the most critical period that expose piglets to stress due to changes in environment, feed, nutrition, and the psychological disruption [9]. Additionally, it is now a great challenge for the swine industry to reduce energy cost and noxious gas emissions from pig house, and also ensure animal welfare [1]. Furthermore, excess $CO_2$, $NH_3$ and $H_2S$ gas emissions in farmhouses has a negative impact on animal's health, and associated workers [7]. A geothermal heat pump can generate energy (electricity) and can be used for cooling or heating the house effectively [10]. The geothermal heat pump was also reported to be a useful heat source for heating poultry houses [6]. This ground source heat pump technology is the most energy efficient, cost effective and environmentally friendly system among heating and cooling systems [11]. On the other hand, sunlight is the largest and cheapest energy resource (solar energy) among the energy systems on the earth [12]. Solar energy has been used in agricultural processes, for example in irrigation, green house, poultry and livestock farming. The establishment of solar renewable energy system can reduce the fuel consumption and enhances the sustainability of agricultural production [5]. However, limited studies were performed with the application of geothermal plus sunlight-based incubator in swine farming. Therefore, this study was focused on the adaptation of geothermal plus sunlight (GS) based incubator for the healthy and economic pig production by evaluating the growth performance and housing environment of piglets.

## 2. Materials and Methods

### 2.1. Animals and Experimental Design

This research was conducted at the experimental swine farm of Sunchon National University, the Republic of Korea. The animals care procedure was approved by the review committee of the Animal Use and Care Council of the university. The trial was performed from 10 June 2022 to 5 August 2022 (8 weeks). Considering the average body weight, $7.7 \pm 0.015$ kg (mean $\pm$ std.), a total of 20 piglets [(Duroc) $\times$ (Landrace White $\times$ Yorkshire)] were randomly divided into two separated incubators while every room had ten individual pens as replications. Each pen was equipped with stainless steel feeder with nipple drinker. The control incubator pens were heated by using 600 W heating lamps while the other room was designed by installing geothermal plus sunlight (GS) heat pump (Figure 1). Piglets were fed commercial diets (data not shown).

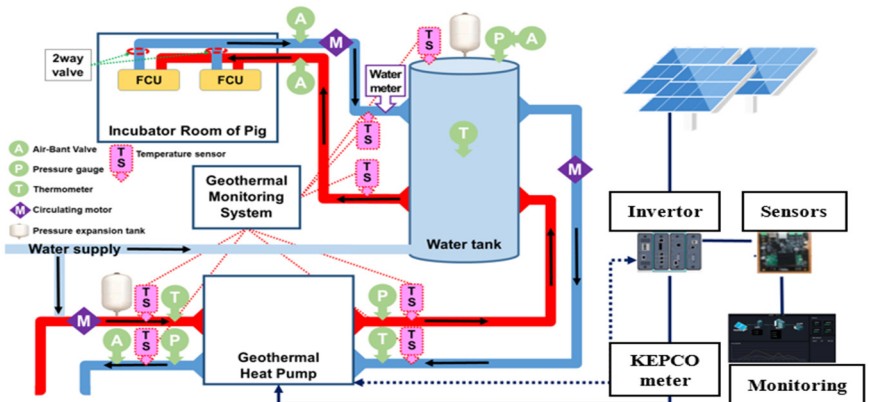

**Figure 1.** Geothermal plus sunlight-based heat pump.

### 2.2. Growth Performance and Feed Cost to Weigh Gain Ratio

At the beginning and at the end of the trial, the weight of the individual piglets was considered as initial and final body weight (kg), respectively. Additionally, data for feed intake and body weight of each pig was measured on a weekly basis. The average daily body weight gain was calculated by subtracting the previous weight from the present weight and divided by the experimental period. The feed was offered twice daily, and feed intake was considered by subtracting the remaining feed from provided feed on a weekly basis. Individual designed feeder was used for each pig. The feed conversion ratio (FCR) was determined by dividing the feed intake by the weight gain of each pig on a weekly basis and the average value was calculated.

Total feed cost was calculated by multiplying the total feed intake with per kg feed price based on South Korean local market price for growing pigs and expressed in USD. Feed cost per unit of body weight gain (cost: gain ratio) was expressed on the ratio of total feed cost and total body weight gain of growing pigs. 1 kg feed price = 1535 South Korean Won and 1535 South Korean Won was equal to 1.07 USD (on 25 October 2022).

### 2.3. Electricity and Housing Environment

The electricity consumption inside the GS incubator and the control incubator were measured by two individual smart energy electrical sub-meters.

The average consumption of electricity = total uses/number of pigs (10);

The reduced electricity consumption = average uses of electricity in the control incubator − the GS incubator;

Saving efficacy = (reduced consumption/avg. uses in control) × 100.

Cost saving = reduced electricity consumption × electricity price.

Temperature and humidity of incubators were measured by sensors with thermocouples and thermistors. The range for temperature and humidity were −2 °C to 80 °C and 0–100%, respectively, and these were were linked with data logger (CR10X Data Logger, Edmonton, AB, Canada) [7]. The concentration for ammonia and carbon dioxide were measured by using a $NH_3$ sensor 3E 100 SE (Bonn, Germany) and a sensor based on bandiburri smart farm monitoring system (NareTrend, Inc. Bucheon City, Republic of Korea), respectively [7].

### 2.4. Statistical Analysis

The trail data were examined with one-way analysis of variance (ANOVA) by SPSS [13]. An individual pig was the replication unit. Data was normally distributed, and homogeneity of variance was performed by levene's test. Duncan's multiple range test was used to assess the differences between means. The results were presented in table as mean ± SEM with the significant level at $p < 0.05$.

## 3. Results and Discussion

### 3.1. Growth Performance and Feed Cost to Weight Gain Ratio

The effect of geothermal plus sunlight (GS) based incubator on feed intake, body weight gain, feed conversion ratio (FCR) and feed cost are shown in Tables 1 and 2, respectively. During the entire 8 weeks of experimental period, no significant differences ($p > 0.05$) were noted on the final body weight, average daily gain (ADG), feed intake and FCR. Daily weight gain, feed intake as well as FCR are the most important indicators of swine production. The adaptation of technology can increase the production that leads to improvement of socioeconomic conditions for farmers [14]. Using GS heating system in the piglet's incubator had no adverse effects on the growth performance of piglets in this study. Moreover, the GS-based incubator had no adverse effect on the daily weight gain and FCR; these were in line with the previous findings with an air heat pump and geothermal heat pump system of pig house, respectively [15,16]. According to Table 2, per unit body weight gain cost (feed cost to weight gain ratio) was lower in the GS-based

incubator than the control incubator of growing pigs, which might improve the economic returns to pig farmers.

**Table 1.** Effects of geothermal plus sunlight-based incubator on growth performance of piglets.

| Parameters | Control Incubator | GS-Based Incubator | SEM | *p* Value |
|---|---|---|---|---|
| Initial body weight (kg) | 7.69 | 7.72 | 0.23 | 0.991 |
| Final body weight (kg) | 42.35 | 45.30 | 1.21 | 0.236 |
| Average daily gain (g) | 618 | 671 | 19.05 | 0.187 |
| Average daily feed intake (g) | 1108 | 1146 | 105.96 | 0.849 |
| Feed conversion ratio (FCR) | 1.78 | 1.70 | 0.063 | 0.449 |

GS, geothermal plus sunlight; number of piglets each group = 10; SEM, pooled standard error of the means; Level of significant ($p < 0.05$).

**Table 2.** Feed cost to weight gain ratio of growing pigs.

| Parameters | Control Incubator | GS-Based Incubator |
|---|---|---|
| Total body weight gain (TBWG, kg) | 34.66 | 37.58 |
| Total feed intake (kg) | 62.05 | 64.18 |
| Total feed cost (TFC, USD) | 66.40 | 68.67 |
| FC: BWG | 1.92 | 1.83 |

GS, geothermal plus sunlight; Feed cost was calculated based on South Korean local market feed price; 1 kg feed price = 1535 South Korean Won; 1535 South Korean Won equal to 1.07 USD (on 25 October 2022).

### 3.2. Electricity Consumption

The energy (electricity) consumption of pig house is shown in Table 3. The total and the average electricity consumption was lower in the GS-based incubator than the control incubator. Compared with the control incubator, the reduced energy consumption (electricity) was 120.95 kWh/8 weeks/head and the energy saving efficiency was 64.76%, respectively. The cost saving from electricity was 3.26 USD/head during the experimental period, and this might ensure a better economic return to farmers. The higher energy saving efficacy in the G- based incubator was due to the uniform distribution of heating with the minimum operating hours by the sunlight. It was reported that the geothermal heating system could provide three units of electricity per unit of consumption, resulting in lower electricity uses than the conventional heating system [17]. In addition, the cost of electricity can be reduced by applying the geothermal heating system in livestock farms [7,18]. Wu [10] also reported that the electricity consumption cost can be reduced by about 46% with air source heating system than the conventional heating system. The price of fossil fuels is increasing continuously.

**Table 3.** Effects of geothermal plus sunlight-based incubator on electricity uses.

| Parameters | Control Incubator | GS-Based Incubator |
|---|---|---|
| Total consumption (kWh/8 weeks) | 1867.68 | 658.10 |
| Average consumption (kWh/head) | 186.76 | 65.81 |
| Compared with control incubator | | |
| Reduced electricity consumption (kWh/8 weeks/head) | | 120.95 |
| Saving efficacy (%) | | 64.76 |
| Cost saving (USD/head) | | 3.26 |

GS, geothermal plus sunlight; cost was estimated according to electricity consumption per day, the current value of 1 kWh = 39.2 South Korean Won (agricultural uses); 39.2 South Korean Won equal to 0.027 USD (on 25 October 2022).

### 3.3. Housing Environment

The effects of the GS-based incubator on the housing environment of piglets is shown in Table 4. The internal temperature and the humidity were not affected by the GS-based incubator ($p > 0.05$) in this study. Importantly, ammonia (ppm) and carbon dioxide (ppm) concentration were lower ($p < 0.05$) in the GS-based incubator than in the conventional control incubator. The temperature and the humidity of pig house are important environmental parameters for good health and production of swine [1]. Variation of temperature inside the pig house may negatively affect the growth performance of pigs [19]. Piglets are usually susceptible to cold stress, especially in the weaning period. In this study, the GS-based heating pump effectively converted the heat from the ground source and the sunlight to the pig barn that ensured a uniform temperature inside the pig barn. The uniform temperature of the house provides a comfortable environment for piglets that ensured the efficiency of the GS-based incubator system for pig farming. In some studies, the average temperature for piglet's house were maintained within 24 to 26 °C and humidity within 60–75%, which were within the range of our current findings [20,21]. Ammonia and carbon dioxide gas are well-known harmful gases to human and animal's health. The emission of ammonia and hydrogen sulfide as harmful gases from swine farm might have negative effects on a pig's growth, health and welfare [22]. The tolerate level of ammonia ($NH_3$) is 20 ppm, according to the international commission of Agriculture and Bio-System Engineering [23]. The growth rate of pigs may fall up to 30% with increased levels of $NH_3$ in the pig house [22]. In this study, both the ammonia and carbon dioxide levels were significantly lower in the GS-based incubator system than the conventional incubator, which can ensure the good health of piglets in the GS-based incubator. Lower harmful gases inside the GS-based incubator may be due to the passage of fresh air in the GS-based incubator. The entry of fresh air may have a role in minimizing or diluting the ammonia, hydrogen sulfide and carbon dioxide gases [15]. Moreover, lower uses of electricity in the GS-based incubator resulted in reducing the level of carbon dioxide inside the house. The lower carbon dioxide emission in the pig house was related to the lower uses of electricity [1]. Saha et al. [24] also reported that installing a pit ventilation system could improve the air quality and reduce the ammonia level about 37–53% in a pig house. According to the previous report, the concentration of $NH_3$ was reduced by about 30% in the geothermal heating system in a pig house [25]. In addition, energy combustion was associated with the sunlight- and geothermal-based incubator system that could reduce the harmful gases in the pig house. Furthermore, the GS heating pump system as source of renewable energy can play role in energy security and may contribute to global climate change. Therefore, the application of geothermal plus sunlight energy can be aneffective solutios for those problems associated with high odor gas emission in the pig house and energy consumption for economic purposes.

**Table 4.** Effects of geothermal plus sunlight-based incubator on environment of pig house.

| Parameters | Control Incubator | GS-Based Incubator | SEM | *p* Value |
|---|---|---|---|---|
| Temperature (C) | 26.42 | 25.12 | 1.18 | 0.975 |
| Humidity (%) | 76 | 74 | 1.25 | 0.954 |
| $CO_2$ (ppm) | 1257 [a] | 917 [b] | 19.98 | 0.045 |
| $NH_3$ (ppm) | 1.86 [a] | 0.59 [b] | 0.224 | 0.041 |

GS, geothermal plus sunlight; Level of significant $p < 0.05$; SEM, pooled standard error of the means; $CO_2$, carbon dioxide; $NH_3$, ammonia. [a,b] values in the same row with different letters are significantly different at $p < 0.05$.

### 4. Conclusions

The uses of the geothermal plus sunlight-based incubator system could reduce the electricity uses, odor gas concentration in a pig house without affecting the normal growth of piglets. Considering the health benefits, electricity consumption and growth performance of pigs, the geothermal plus sunlight-based incubator can be adapted as a renewable source of energy for the sustainable pig farming.

**Author Contributions:** Conceptualization, C.-J.Y.; writing original daft preparation, S.M. and H.-S.M.; Preparation of Manuscript, S.M., H.-S.M. and M.A.D.; Editing and revised, S.M., K.M.B.A., V.C., Y.-H.K., J.-P.M. and C.-J.Y.; Supervision, C.-J.Y. All authors have read and agreed to the published version of the manuscript.

**Funding:** This research received no external funding.

**Institutional Review Board Statement:** The animals care procedure was approved by the review committee of the Animal Use and Care Council of Sunchon National University (SCNU IACUC-2021-04).

**Informed Consent Statement:** Not applicable.

**Data Availability Statement:** Not applicable.

**Acknowledgments:** This work was carried out with the support of "Cooperative Research Program for Agriculture Science & Technology Development (Project title: Development of Combined Heat Sources Heat Pump Utilization Technology Using Renewable Energy, Project No. PJ014967022022)" Rural Development Administration, Republic of Korea. This paper was supported by (in part) Sunchon National University Research Fund in 2020 (Grant number: 2020-0209).

**Conflicts of Interest:** The authors declare no conflict of interest.

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
