# Peer review of "Geothermal Plus Sunlight-Based Incubator for Sustainable Pig Production"

_sustainability, doi:10.3390/su142215243_

Round 1

Reviewer 1 Report

Taking into concideration the energy problem worldwide, the paper is of great interest. First of all, the results are well expected, meaning that if you use an alternative source of energy, that will decrease electrical energy consumpion. So, the results are logical and as expected. You should also answer what will be the investment cost and how the producers, facing energy and feedstufs cost can invest in such technologies. There are many, in my point of view, misues of literature references, speaking about statements that i could not find in them. I have made some comment in the attached file, which i hope that will help you. I would like also to see an analysis about the efficacy of making such an investemet and when the producer will get his money back. 

Author Response

Reviewer -1

Taking into consideration the energy problem worldwide, the paper is of great interest. First of all, the results are well expected, meaning that if you use an alternative source of energy, that will decrease electrical energy consumption. So, the results are logical and as expected. You should also answer what will be the investment cost and how the producers, facing energy and feedstuffs cost can invest in such technologies. There are many, in my point of view, misuse of literature references, speaking about statements that i could not find in them. I have made some comment in the attached file, which i hope that will help you. I would like also to see an analysis about the efficacy of making such an investment and when the producer will get his money back

Authors Response: Dear Reviewer-1, Thank you very much for your good advises and comments. As per your advises, we have considered the feed cost, electricity cost, and the cost involved in per unit of body weight gain of growing pigs in this study. We have added new table in text.

Please check the table no. 2 and table 3. Thank you. 

Line no. L35 (Ref. 3); 100-104,113;138-141;155-156.

This trail is related only growing pigs. So we could not calculate the market price of finishers pigs (meat) for economic returns (money back). But we hypothesized that the lower maintained (electricity) cost must be profitable to the pig farmers. In addition, it is difficult to calculate the full returns from a single trail against the total investment specially the Geothermal and Solar energy system as infrastructure investment. But we believe it must be economic for long time application.  

In addition, thank you very much for your comments regarding the statement of cited References in text. Now we have revised all cited references and mentioned appropriately in text.

Please check the Line no. L35; 49; 52-57; 133-134;179-183;195-196;

We have replaced two references (ref no, 4 and 14) in text and in references list.

We have also revised all the suggested minor writing errors and language issue, in text with color for your kind consideration.

Thank you very much for your time and necessary comments to improve our submission.

Thank you.

Reviewer 2 Report

The aim of this study was to assess the effects of geothermal plus sunlight based incubator on the growth performance, electricity uses and housing environment of piglets.

The study is interesting and solidly written, but I would have a few suggestions and remarks.

If the p values are greater than 0.05, avoid interpreting the numerically larger value and the like (lines 28-29, 114-117). In that case, there are no differences.

It is not clearly described how you calculated ADFI and FCR. If it is a group average, how did you test for differences? If it is a matter of individual data per pig, it is necessary to explain how you got that data. Actually, how was the amount of feed consumed by each pig during the experimental period recorded/calculated. This should be explained in subchapter 2.2.

If you applied ANOVA, was homogeneity of variance tested beforehand? If so, by what test? This should be stated in subchapter 2.4.

Reference 20 is incomplete (missing publication year).

Author Response

Reviewer-2

The aim of this study was to assess the effects of geothermal plus sunlight based incubator on the growth performance, electricity uses and housing environment of piglets.

The study is interesting and solidly written, but I would have a few suggestions and remarks.

If the p values are greater than 0.05, avoid interpreting the numerically larger value and the like (lines 28-29, 114-117). In that case, there are no differences.

Authors Response: Dear Reviewer-2, Thank you very much for your time and good advises to improve our manuscript. We have revised our submission as per your advises.

We have deleted the related statement from text.

It is not clearly described how you calculated ADFI and FCR. If it is a group average, how did you test for differences? If it is a matter of individual data per pig, it is necessary to explain how you got that data. Actually, how was the amount of feed consumed by each pig during the experimental period recorded/calculated. This should be explained in subchapter 2.2.

Authors Response: We have added the information in text. Thank you.

Please check the line number 81-83; 92-99

If you applied ANOVA, was homogeneity of variance tested beforehand? If so, by what test? This should be stated in subchapter 2.4.

Authors Response: We have added the information in text. Thank you.

 Please check the line number 122-124

All other changes have been made in color for your kind consideration.

Hope you will consider our revised submission and many thanks for your valuable time for us.

Thank you.

Round 2

Author Response

Reviewer -1 (Round 2)

(From the attached file)

Authors Response: Dear Reviewer-1, Thank you very much for your valuable time and further suggestions on minor writing errors. We have revised our manuscript accordingly in text

Please check the line no: 55-57; 65-66; 70; 102;156. In addition, we have also revised few minor writing errors in text (with color) for your kind consideration. Please check the table 3, table 4 and Ref. no. 3.

Thank you very much.

Reviewer 2 Report

The revised version of the manuscript has been greatly improved.

Author Response

Reviewer-2 (Round 2)

Reviewer comments: The revised version of the manuscript has been greatly improved.

Authors Response: Thank you very much to satisfy with our revised submission.

Many Thanks for your valuable time.